# Laser Light as an Emerging Method for Sustainable Food Processing, Packaging, and Testing

**DOI:** 10.3390/foods12162983

**Published:** 2023-08-08

**Authors:** Prasad Chavan, Rahul Yadav, Pallavi Sharma, Amit K. Jaiswal

**Affiliations:** 1Department of Food Technology and Nutrition, Lovely Professional University, Phagwara 144402, India; erprasad.chavan@gmail.com; 2ICAR-Directorate of Floricultural Research, Pune 411036, India; rahul.yadav@icar.gov.in; 3Quality Management Officer, Fresh Company GmbH, 71384 Weinstadt, Germany; psharma9136@yahoo.com; 4School of Food Science and Environmental Health, College of Sciences and Health, Technological University Dublin, City Campus, Central Quad, Grangegorman, D07 ADY7 Dublin, Ireland; 5Environmental Sustainability and Health Institute (ESHI), School of Food Science and Environmental Health, Technological University Dublin, City Campus, Grangegorman, D07 H6K8 Dublin, Ireland

**Keywords:** non-destructive testing, laser ablation, food packaging, microbial inactivation, backscattering imaging

## Abstract

In this review article, we systematically investigated the diverse applications of laser technology within the sphere of food processing, encompassing techniques such as laser ablation, microbial inactivation, state-of-the-art food packaging, and non-destructive testing. With a detailed exploration, we assess the utility of laser ablation for the removal of surface contaminants from foodstuffs, while also noting the potential financial and safety implications of its implementation on an industrial scale. Microbial inactivation by laser shows promise for reducing the microbial load on food surfaces, although concerns have been raised about potential damage to the physio-characteristics of some fruits. Laser-based packaging techniques, such as laser perforation and laser transmission welding, offer eco-friendly alternatives to traditional packaging methods and can extend the shelf life of perishable goods. Despite the limitations, laser technology shows great promise in the food industry and has the potential to revolutionize food processing, packaging, and testing. Future research needs to focus on optimizing laser equipment, addressing limitations, and developing mathematical models to enhance the technology’s uses.

## 1. Introduction

Fresh agricultural food products, including meat, fruits, and vegetables, serve as the primary source of essential nutrients for maintaining human health. Ensuring the safety and quality of these agricultural products throughout the farm-to-fork process is of utmost importance. Fresh products are prone to spoilage and deterioration due to physiological, biological, biochemical, and microbiological factors, leading to a decline in their quality [1]. Various techniques have been developed to extend the shelf life of food products. Although heat processing has proven effective in eliminating microbes, it may also lead to undesirable changes in the physical, sensory, and nutritional properties of food products. These limitations and market demand have prompted the development of innovative technologies [2,3,4].

Food preservation techniques play a vital role in prolonging the shelf life of products. However, traditional processing methods often degrade bioactive compounds and can be destructive. As a result, researchers are increasingly focusing on the development of novel, green technologies that are efficient, user-friendly, affordable, and non-destructive. Among these emerging technologies, laser technology has gained recognition as an effective, efficient, and contactless approach, offering several advantages over conventional methods.

The application of laser technology in the food industry is a relatively new concept, with potential uses in processing, preservation, and packaging. The acronym “LASER” stands for “Light Amplification by Stimulated Emission of Radiation,” a process in which light emitted by a laser source differs significantly from ordinary light [5]. Laser light consists of electromagnetic energy with wavelengths ranging from 1 nm to 1800 nm. When atoms absorb energy, their electrons transition to higher energy levels, subsequently releasing excess energy as photons when they return to lower energy levels. This process produces a photon beam or laser light [6]. In contrast, ordinary white light emits a combination of isotropic (omnidirectional) light with various wavelengths, whereas laser light is bright, coherent, monochromatic, and unidirectional. Food is quickly cooked by carbon dioxide (CO_2_) lasers, which are utilized in a variety of food preparations [7]. The primary use of laser heating and cooking, which combines electromagnetic radiation and laser beams, is in the 3D food printing industry for the assembly of multilayer foods. The lasers may be utilized to produce micromoles on plastic film at the rate of a commercial film manufacturing line compared to the needle puncture method. The photochemical or photothermal process, which is determined by wavelengths and the molecular composition of the material, accounts for how the laser interacts with the materials [8]. The dough was previously baked with lasers, and blue lasers can gelatinize starch but prohibit browning [9]. The mechanics and properties of laser beam technology are covered in this article, along with how laser light is used in food processing such as microbial inactivation, laser-assisted cooking, laser ablation, meat marination, and food packaging. Each application of laser is explained in detail under different sections. Additionally, the application of laser beam technology to food product packaging and testing were discussed.

## 2. Properties of Laser Light

*Monochromatic*: A light wave characterized by a single vibration frequency is considered monochromatic. While no light source is perfectly monochromatic, laser light comes exceptionally close, with minimal deviation due to Doppler shifts arising from the motion of photon atoms within the medium. The spread in frequency (∆ν) and wavelength (∆λ) quantitatively measures monochromaticity. The narrow bandwidth of the wavelength emitted by a laser source enables precise targeting within tissue, minimizing damage to adjacent structures.

*Coherent*: Coherent light waves are those that travel “in-phase” with one another, meaning that their peaks and troughs align precisely. In contrast, ordinary light sources, such as incandescent and LED, emit light with random phases. Coherence measures the accuracy of the waveform. Stimulated photons emitted by the laser source, for instance, are in step with stimulating photons.

*Brightness*: Lasers emit light that is significantly brighter and more intense than conventional light sources. Brightness is measured as the power emitted per unit area per unit solid angle. For example, a 1 mW He-Ne laser (2.4 × 10^12^ W/m^2^ sr^1^) is 10⁶ times brighter than the sun (1.3 × 10^6^ W/m^2^ sr^1^).

*Unidirectional*: Light emitted from a laser travels in a highly collimated, narrow beam with minimal divergence. This property is attributed to the precise alignment of parallel mirrors that form an optical cavity. The unidirectional nature of laser light allows it to focus on a very small spot.

The intensity of laser light: Power intensity is another crucial factor for photon energy absorption. Increasing power densities can address the issue of insufficient energy absorption by materials at a specific wavelength [9]. At standard power levels, materials’ energy absorption is linear and consistent with Beer–Lambert’s law. However, at extremely high intensities, energy absorption becomes nonlinear (absorption coefficients of materials change with power intensity), and a plasma plume forms where the laser interacts with the substrate [6].

The intensity of laser light (irradiance) is calculated by Equation (1):I = P/A (W/m^2^)(1)
where P is the power of the laser beam, which represents the rate at which optical energy is delivered by the beam and is measured in J/sec or W. As the shape of the laser beam is circular, the cross-sectional area of the beam, A, can be calculated by Equation (2):A = π(d/2)^2^(2)
where d is the diameter of the beam. Thus, the intensity of laser light can be expressed as Equation (3):I = P/(π(d/2)^2^)(3)

These laser properties have found extensive applications in various fields, including military, metallurgy, biomedical, biochemistry, sanitation, and pasteurization, among others.

## 3. Mechanism of Laser Light-Induced Food Processing

Thermal and photochemical processes form the basis of the fundamental laser-assisted food processing concept. The following explanation provides an insight into the thermal process: when a laser beam interacts with a substrate’s surface, the laser photons’ energy is absorbed by the electrons of atoms present in the substrate. This absorbed energy is then converted into heat energy, causing the substrate to heat up, in accordance with Beer–Lambert’s law. As a concentrated laser beam interacts with the food substrate’s surface, its photons’ energy is absorbed by the electrons of atoms within the substrate [10]. Beer–Lambert’s law states that the absorbed energy transforms into heat energy, which subsequently heats the substrate. Additionally, the extent of light absorption is influenced by the thickness of the materials and the intensity of the light source.

As the temperature continues to rise due to the influx of photon energy, the thermal effect induces melting or vaporization in the substrate. This process ultimately results in the removal of macroscopic materials from the substrate [11]. The transition from a solid state to a gaseous state led to the formation of a plasma plume [10,11]. The vaporization of materials generates local high pressure, also known as recoil pressure, which propels molten materials out of the laser–substrate interaction zone and into the ejected region. These ejected materials may redeposit on the substrate, altering its energy absorption capacity [7,12].

During a photochemical reaction, molecules present in the substrate can break their chemical bonds by absorbing high photon energy. This process initiates or activates several chemical reactions, including radical formation, crosslinking, and chain scission [5]. The primary mechanism of laser-assisted food processing may be photochemical, thermal, or a combination of both. Figure 1 illustrates this mechanism, which is determined by a combination of material properties and specific laser parameters.

## 4. Application of Laser Irradiation 

The rapidly evolving field of laser-based technology showcases the potential for enhancing the quality and safety of food products. There is a growing body of evidence supporting the application of laser light in various food processing operations, such as microbial inactivation, food drying, laser ablation, and meat marination. In addition to these applications, numerous studies have been conducted in the food industry due to the continuous advancements in laser technology, particularly in the areas of food packaging and detection. Examples of such applications include laser-assisted surface modifications of polymers to create functional packaging materials and laser-induced microporous-modified atmosphere packaging [10,13,14].

### 4.1. Laser Pretreatment 

Laser pretreatment technology enables precise material modification using a flexible, non-contact method, significantly reducing the risk of microbiological contamination associated with equipment [15]. Different parts of raw materials absorb laser energy at varying rates, allowing selective processing compared to conventional methods. The millimeter-sized focal point created by a laser beam is ideal for localized pretreatment, sparing the rest of the raw materials from damage and minimizing waste. Moreover, non-contact laser pretreatment enhances processing accuracy and efficiency while decreasing the likelihood of microbial and chemical contamination [6,16].

Fujimaru et al. [17] used a CO_2_ laser to create perforations on the skin of frozen blueberries, enhancing sugar infusion during osmotic treatment. By increasing perforation density and depth, solute transport into the fruit was facilitated, resulting in higher fruit weight with up to a 65.34% increase at 30° Brix/d solute concentration. Munzenmayer et al. [18] similarly employed a CO_2_ laser to perforate blueberry skin, improving mass transfer during the freeze-drying process. With nine micro-perforations per berry, primary drying time was significantly reduced from 17 ± 0.9 h for untreated berries to 13 ± 2.0 h, while the percentage of non-burst blueberries increased from an average of 47% to 86%, indicating a significant quality improvement.

Teng et al. [6] combined CO_2_ laser pretreatment with ultrasound before infrared-freeze-drying of blueberries, resulting in reduced fruit substance (e.g., sugar and organic acids) exposure outside the epidermis. Qu et al. [14] also performed CO_2_ laser pretreatment on raspberries before pulse-spouted microwave freeze drying, creating 0, 12, 24, and 36 holes per fruit. Consistent with Munzenmayer et al. [18], the drying time decreased by up to 23.08% as the number of perforations per berry increased. Moreover, the reduced drying time positively impacted product quality, lowering the shrinkage rate by 3.95% and increasing anthocyanin retention by 20.02%. The drying process improvement is attributed to the perforations allowing vapor generated within the berry bulk to escape unimpeded by the waxy skin.

Laser pretreatment is well known in extraction techniques due to its excellent ability to disintegrate cellular membranes without harming cell factory compartments or other internal compounds. The most significant benefits of laser treatment include its simplicity, speed, and lack of organic solvent use [19,20]. In lipid extraction from oleaginous microorganisms, various pretreatment techniques were employed to disrupt oleaginous microalgae *Nannochloropsis oculata* cells, including microwave, water bath, blender, ultrasonic, and laser treatment. Laser treatment proved most effective at cell disruption (96.53%), followed by microwave treatment (94.92%) [21]. However, research on using laser pretreatment for bioactive compound extraction from plant materials, flowers, and other organic matter is limited. There is a need for extensive research on optimizing laser-assisted hybrid techniques to study the laser’s effect on extraction efficiency and compound quality, as well as the techno-economic feasibility of the technique.

### 4.2. Microbial Inactivation

Non-thermal methods such as ultrasonication, high-pressure processing (HPP), irradiation, UV light, pulse electric field (PEF), and electromagnetic field are being researched for microbial inactivation. However, these methods can be expensive, less energy efficient, and pose risks for operators. Spores of Bacillus and Clostridium species are resistant to dry heat, desiccation, UV, irradiation, high pressure, oxidative chemicals, and even simulated meteor impact [22,23].

#### Mechanism of Microbial Inactivation

Microbial inactivation through laser irradiation involves directly irradiating food materials with laser light, typically passing the laser beam through a beam extender (Figure 2). The inactivation mechanism depends on the wavelength, continuous or pulsed wave, and pulse rate. Laser beams are generally characterized by their wavelength, output power or energy, and beam diameter. The application of a continuous-wave femtosecond (fs: 10−15 s) laser in the visible or infrared (400–800 nm) region coherently excites mechanical vibrations in protein capsids of targeted viruses, causing damage and inactivation of a broad range of viruses and bacteria. Conversely, low-power nanosecond visible pulsed lasers inactivate bacteria based on the transient photothermal evaporation effect. Bacterial strains photothermally evaporate upon instantaneous absorption of pulsed irradiation. Kohmura et al. [24] achieved a three-log inactivation of *Escherichia coli* in 10 min using a low irradiation dose of 50 kJ/cm^2^.

Medical scientists at Arizona State University have demonstrated the destruction of viruses and bacteria such as HIV by applying femtosecond (fs: 10−15 s) infrared laser pulses of a selected wavelength and pulse width through the Impulsive Stimulated Raman Scattering (ISRS) process. This method produces lethal vibrations in the protein coat of microorganisms, destroying them without damaging human cells (Institute of Physics, 2007).

Maktabi et al. [26] found that the bacteriostatic effect of Nd: YAG laser treatment (ƛ = 1064 nm) on *Escherichia coli* and *Pseudomonas fragi* increased by 1.50–2.68 and 1.31–2.50 log CFU/mL, respectively, compared to conventional ultraviolet light (ƛ = 180–280 nm). In addition, they used diode laser irradiation at 450 nm wavelength and 1.3 mW/cm^2^ light intensity for 3, 6, and 12 min to preserve the quality characteristics of strawberry fruit during cold storage. The results showed a significant decrease in weight loss and decay percentages of strawberries treated with laser light for 3 and 6 min compared to untreated strawberries, with the laser treatments preserving flesh firmness until the fifth day of storage. Even with the longest laser exposure time, laser irradiation significantly decreased the reduction percentage of antioxidant activity compared to the control sample.

Gonca et al. [27] used a diode laser to inactivate Gram-positive and Gram-negative bacteria and yeast, and to disinfect wastewater and natural milk. The researchers investigated the effect of diode laser (450 nm multimode semiconductor diode) processing parameters, such as laser type, power density, irradiation time, laser penetration efficiency, and biofilm inhibition on pathogenic microorganisms such as *Escherichia coli* (ATCC 10536), *Staphylococcus aureus* (ATCC 6538), and *Candida albicans*. The blue laser was found to be more efficient than red and green lasers. At a power density of 0.36 W/cm^2^ for 15 min, the rates of inhibition for *S. aureus*, *E. coli*, and *C. albicans* were 65.9%, 34.52%, and 43.63%, respectively. After 30 min of blue laser irradiation, the microbial growth inhibitions for *S. aureus*, *E. coli*, and *C. albicans* were determined to be 85.39%, 41.18%, and 54.55%, respectively. The greatest biofilm inhibition for *S. aureus* (94.61%) was observed when cells were subjected to blue laser irradiation for 60 min. Additionally, wastewater and natural milk were successfully sterilized by blue laser irradiation at a laser power density of 0.54 W/cm^2^. 

The application of laser irradiation for microbial inhibition in different food materials has been summarized by the authors of [6]. According to the available literature, the coherent and monochromatic light of lasers reduces damage to other food components, while their high peak power pulses have a better ability to penetrate turbid and semi-turbid liquids. Numerous studies on seeds have demonstrated that laser irradiation technology has the potential to be a promising alternative strategy to inhibit the growth of decay-causing pathogenic microorganisms. The results indicate that lasers have a more significant bacteriostatic effect than conventional ultraviolet radiation (Table 1).

### 4.3. Laser-Assisted Cooking

Food Layered Manufacture (FLM) requires precise and regulated heating to ensure the taste and appearance of the product. Laser cooking differs from traditional cooking methods as it offers more precise control over the timing and location of heat application [34]. The laser heating mechanism allows high-resolution food processing, which can be parameterized by factors such as speed, power, and spot size [9]. Three-dimensional printing technology can create foods with intricate geometric designs or specialized textures and nutrients. However, a significant challenge is the unfavorable deformation, such as expansion, that printed foods experience during post-treatment [35].

Fukuchi et al. [36] proposed a novel culinary approach of dry heating using a laser cutter. This technique cooks ingredients according to their shape and composition using a computer-controlled laser cutter and a video image-processing technique, creating unique flavors, textures, decorations, and distinctive identifiers on the ingredients. A team from Columbia University conducted extensive research to modulate the operational parameters (cooking patterns, flux, power, speed, repetition rate, etc.) of CO_2_ lasers and blue lasers to cook dough and Atlantic salmon, developing a laser-induced visual model for dough browning to investigate the feasibility of laser cooking printed products [9,34,37,38].

The researchers found that by modifying the dough’s water content and the laser’s exposure pattern, it is possible to control the temperature and depth of heat penetration when using a high-resolution blue diode laser operating at 445 nm. Compared to an infrared laser, a blue laser significantly increased the amount of heat that could be penetrated into dough products. Combining a blue laser with an infrared laser produces optimal cooking conditions for food layers [9]. A data-driven model was developed and validated, generating photorealistic RGB images of dough surface browning using a CO_2_ laser dough browning pipeline. This model demonstrated the ability to characterize the visual appearance of browned samples, such as surface color and patterns. The model outputs a 64 × 64-pixel image of laser-browned dough based on laser speed, energy flux, and dough moisture, aiding in designing laser-browned objects with pre-designed patterns and browning levels [38].

Selective laser broiling, a unique food processing method, employs a two-axis mirror galvanometer system to direct laser power to cook raw food. A trochoidal scanning pattern optimized Atlantic salmon frying. The color analysis examined how the trochoidal cooking pattern’s geometry (circle diameter, circle density, and period), heat flux (2.71 MW/m^2^ and 0.73 MW/m^2^), and power (2 W and 5 W) of the blue laser affected the interior temperature and heat penetration. Blue laser heat at 445 nm penetrated fish up to 2 mm, denaturing proteins for frying thin food layers [37]. In another study, Blutinger et al. [39] developed a precision cooking method for printed foods using multiwavelength lasers through customized software-driven patterns. Researchers used chicken as a model food and combined the cooking capabilities of three different lasers: a blue laser (ƛ = 445 nm), a near-infrared (NIR) laser (ƛ = 980 nm), and a mid-infrared (MIR) laser (ƛ = 106 µm). They discovered that IR light browns more effectively than blue light, NIR light can brown and cook foods through packaging, and laser-cooked foods experience about 50% less cooking loss than foods broiled in an oven. Integrating software into the cooking process will promote more innovative food design, allowing individuals to precisely personalize their meals and create new markets for this expanding sector.

### 4.4. Laser Ablation

Laser ablation, also known as photoablation, involves removing material from a solid (or occasionally liquid) surface by irradiating it with a laser beam. The material absorbs laser energy, heats up, and evaporates or sublimates at low laser ranges. Laser ablation typically refers to the removal of material using a pulsed laser, although it can be achieved with a continuous-wave laser beam if the laser intensity is high enough. While longer laser pulses (e.g., nanosecond pulses) can heat and thermally alter or damage the processed material, ultrashort laser pulses (e.g., femtoseconds) cause minimal material damage during processing due to the ultrashort light–matter interaction, making them suitable for micromaterial processing [10]. 

Ornela et al. [40] developed a machine for de-thorning nopal using a pulsed Nd:YAG laser with specific pulse energy, rate, duration, and laser spot diameter. The sensor detects the presence and location of thorns, then commands their removal with laser pulses. The process takes about 1 min to remove 100% of thorns from each leaf, and microbiological analysis shows improvements in the product’s quality. Mineral composition analysis before and after laser irradiation did not reveal any changes.

Laser treatment for peeling fruits and vegetables has potential applications for food processing enterprises and fast-food establishments. Panchev et al. [41] utilized a CO_2_ laser ablation technique for peeling oranges and lemons before extracting pectin using an HCl water solution. The laser ablation effectively removed peels while maintaining organoleptic qualities such as texture, freshness, and naturalness. They also observed that laser pre-treatment improved pectin yield, gel strength, and purity without significantly affecting molecular weight or esterification.

Ferraz et al. [16] developed a mathematical model for determining the necessary laser power and travel speed to achieve a given cut depth in potato tuber slabs. The model takes into account material characteristics and laser processing factors, with preliminary calculations and trials demonstrating accurate predictions of cut depth by a CO_2_ laser in potato tuber parenchyma. Beldjilali et al. [42] investigated plasmas produced by laser ablation of fresh potatoes using infrared nanosecond laser radiation. Their research indicated that plasma produced by double pulses has a larger volume and a lower density.

Laser ablation has also been used as an alternative to traditional methods for producing edible films, yielding lasting antimicrobial effects. CO_2_ laser treatment on fruits and vegetables released aroma substances that could be captured and utilized. Picca et al. [43] synthesized Cu/Ag nanocolloids of chitosan-based aqueous solutions using a flow cell by laser ablation, intending to use the nanoparticles as antibacterial food packaging materials.

Augusto et al. [44] employed laser ablation for food monitoring using laser ablation inductively coupled plasma-mass spectrometry (LA-ICPMS). Calibration strategies allowed for the determination of various elements in food samples, both liquid (e.g., orange juice) and solid (e.g., dietary supplements). LA-ICP-MS offers advantages such as operational simplicity, versatility, and high analytical capability for multielement determinations.

However, laser ablation has significant disadvantages compared to other preparatory techniques. The high cost of the laser system and the large amount of energy required are notable drawbacks. Moreover, a significant number of nanoparticles positioned along the laser beam cause ablation efficiency to decline with prolonged ablation times, which can be addressed with the careful selection of fluidics. Despite these challenges, researchers have explored ways to improve laser ablation methods for various applications in the food industry. Streubel et al. [45] demonstrated the production of nanocolloids with strict composition control and continuous multi-gram ablation rates (up to 4 g/h) for several metals. They utilized a 500 W picosecond laser source operating at a 10 MHz repetition rate and fully coordinated with a polygon scanner to achieve scanning speeds of up to 500 m/s. This technical approach enables the spatial bypassing of cavitation bubbles that impede higher ablation rates at MHz repetition rates due to the shielding effect.

Laser-ablated silver nanoparticles (AgNPs) have potential applications in food, where it is essential to use non-toxic chemicals. However, the concentration obtained by laser-generated AgNP colloids is often low, making it challenging to implement them on an industrial scale. Sportelli et al. [46] developed a method for preparing stable Ag colloids in pure solvents without using either capping and stabilizing agents or reductants. This technical solution allows spatial bypassing of the laser-induced cavitation bubbles, preventing higher ablation rates at an MHz repetition rate due to the shielding effect.

### 4.5. Meat Marination 

Meat marination is a process widely employed in the meat industry to enhance flavor, tenderness, and juiciness. For example, marinating chicken meat results in a product with improved sensory attributes and shelf life compared to unmarinated chicken. However, achieving a specific salt content in the meat can be time consuming. Laser micro-perforation, a pre-treatment in which the laser creates micropores in the meat, can be combined with vacuum impregnation to expedite the pork meat marination process. Utilizing both technologies can reduce processing time by nearly 48%, significantly increasing plant productivity.

Ramírez et al. [47] applied laser micro-perforation and vacuum impregnation to accelerate poultry meat marination, reducing the processing time by 6 h (approximately 34%) compared to the control, thus substantially enhancing plant productivity. The study employed CO_2_ laser micro-perforation alongside vacuum impregnation on chicken breasts, mathematically analyzing marinating time using Fick’s second law and an anomalous diffusion model. Unmarinated chicken flesh cylindrical pieces were CO_2_-laser microperforated with pores of 228 µm and marinated for 60 h at 6 °C with sodium tripolyphosphate (1% *wt/wt*) and NaCl (3% *wt/wt*) under vacuum pressure (15 kPa). The chicken-to-brine ratio was 1:11 wt/wt. Combining micro-perforation with vacuum pulses promoted marinade diffusion into chicken slices, reducing processing time by 34%. For Fick’s second law, the Deff ranged between 1.46 × 10^−10^ and 2.08 × 10^−10^ m^2^/s, while for the anomalous diffusion model, it ranged between 2.27 × 10^−10^ and 4.23 × 10^−10^ m^2^/sα, with values near 1.

Figueroa et al. [48] also investigated the effects of vacuum impregnation (VI) and CO_2_-laser micro-perforation on pork. A 1500 mm focused 100 W CO_2_ laser tool connected to a computer interface created microperforations, operating at a frequency of 10 kHz and a continuous wavelength of 10.6 µm. Pork cylinders were microperforated and marinated for 60 h at 6 °C in a solution containing 8% *w*/*w* NaCl and 0.3% *w*/*w* Na_5_P_3_O_10_. The increased porosity from the laser allowed the solution to migrate and be retained more easily within the pork slices. When compared to traditional salting methods, the marination technique incorporating VI and microperforations significantly accelerated mass transfer, reducing the marinating processing time by 47.8%.

While the effect of laser-assisted marination on marinating time, mass transfer mechanism, and salt content has been investigated, further research is needed to understand the technique’s impact on organoleptic properties, texture profile, and chemical constituents. Moreover, other types of laser beams, such as blue lasers and Nd:YAG, could be explored in conjunction with other novel advancements, including ultrasonication, high pressure, and moderate electric fields.

### 4.6. Food Packaging

Food packaging plays a crucial role in the complex journey from grower to consumer, as it is an essential food processing operation. Proper packaging can extend the storage life of food products by reducing moisture loss. Modern food packaging often needs to do more than just protect the product. Lasers are currently employed in various scientific and technological fields and have numerous applications in food processing operations, including the food packaging industry. Examples of these tasks include perforating and slitting paper and plastic films. The packaging industry has greatly benefited from laser technology, as lasers are now used for creating micro-perforations on packaging films, welding packaging films, modifying surface structures of packaging materials for fresh produce applications, and scoring packaging materials through laser ablation.

Excimer lasers (248 nm), which are pulsed gas lasers that emit light, can be used as an alternative to chemicals for cleaning and disinfecting surfaces. Pulsed laser beams have effectively removed and killed adhering microorganisms without causing noticeable surface damage. Perishable fresh food materials can be stored in laser-perforated monolayer films to enhance their shelf life and facilitate package opening. Suhem et al. [32] investigated the antifungal activity of michelia oil (*Michelia alba*) and its main components, linalool and caryophyllene, at concentrations ranging from 300 to 500 mg g^−1^ against the growth of *Aspergillus niger*, *Aspergillus flavus*, *Penicillium* sp., *Rhizopus* sp., *Fusarium* sp., and *Cladosporium* sp. on bamboo paper packaging surfaces, both with and without laser treatment. A helium-neon (He-Ne) laser was used for 1 min at wavelengths of 543, 594, 604, 612 and 633 nm to enhance the antifungal activity of these mixtures. Each mold’s spore suspension (7 log10 CFU mL^−1^) was inoculated on the treated bamboo paper.

When laser treatment was applied, michelia oil and linalool exhibited minimum inhibitory concentration (MIC) values of 300 mg g^−1^ and 150 mg g^−1^, respectively. No MICs were identified at the highest concentration (500 mg g^−1^) without laser treatment. The MICs demonstrated the potential of laser-treated bamboo paper boxes with michelia oil and linalool to prevent mold growth on snack bars for at least 5–7 weeks under accelerated storage conditions (25 °C and 100% RH) compared to the control (3 days) [48]. The applications of lasers in food packaging have been summarized in Table 2.

### 4.7. Laser Perforation

Laser perforation is an innovative application of laser technology that facilitates easier package opening without the need for blades or scissors. Laser micro-perforation is used to create breathable packaging, such as modified packaging materials, in order to extend the shelf life of perishable items. The successful application of laser micro-perforation technology in bio-plastics broadens their possibilities for developing optimal equilibrium modified atmosphere packaging (EMAP) systems tailored to the specific requirements of high-value horticultural products [57]. The use of modified atmosphere packaging (MAP) for extending the shelf life of fresh produce is becoming increasingly popular, resulting in potential scope for developing EMAP systems using polylactic acid (PLA) films.

Briassoulis et al. [58] developed an EMAP system using a multilayer structure of laser micro-perforated 30 µm PLA/non-perforated Mater-Bi^®^ sheets, designed to provide appropriate barrier properties for cherry tomatoes and peaches. Peelman et al. [59] investigated the shelf life of rump steak, ham sausage, fillet desaxe, grated cheese, and pre-fried fries that had been MAP-packed in multilayer trays using commercial cellulose-based film/PLA and/or paper/AlOx/PLA films. Bioplastics, particularly PLA, have demonstrated versatility to date, allowing for more precise design and management of the in-package environment compared to conventional materials, which possess higher permeability to water vapor and weaker barrier properties against CO_2_ and O_2_. The successful application of laser micro-perforation technology in bioplastics broadens their potential for developing optimal EMAP systems that cater to the unique needs of high-value horticultural products [57].

### 4.8. Laser Transmission Welding

Laser transmission welding is a non-contact, versatile process for joining thin films of the same or different packaging materials. A bond is formed when laser energy is absorbed and converted into heat, allowing the overlapped materials to be welded without using chemical solvents or adhesives. Poly Lactic Acid (PLA) is an environmentally friendly and versatile polymer that can replace non-biodegradable plastics in food packaging. Pagano et al. [49] used a 20 W fiber laser with a wavelength of 1.64 µm to join commercial PLA of 75 µm thickness with aluminum foil of 25 µm thickness for food packaging. The results showed that the power level of the laser source influenced the joint quality, and the k factor (energy delivered per unit length) helped determine the process’s viability [49]. 

Various packaging materials are employed to extend the shelf life of food products, and laser technology offers valuable benefits to the packaging industry. Even though PLA currently has a low market volume, its production is expected to grow rapidly due to ongoing research in the field. In the experiment conducted by [49], the pulse duration and pulse repetition frequency were kept constant, while the average power and welding speed were varied during the welding operation. The pressure was applied using poly methyl methacrylate in the laser working zone, ensuring close contact between the films.

The quality of the welded line was analyzed by examining the common area using optical microscopy to identify defects and measure the weld dimensions. Bond strength and tensile strength were also observed. The experimental measurements were correlated with the k factor, which is the product of pulse duration, pulse repetition frequency, and the average power ratio to welding speed. The results demonstrated that the joint quality depended on the power level of the laser source, and the k factor aided in understanding the feasibility of the process. Laser perforation is an emerging application of laser technology, enabling the creation of easy-to-open packaging without the need for knives or scissors. Laser micro-perforation is used to develop breathable packaging, such as modified packaging materials, for extending the shelf life of perishable items.

### 4.9. Laser Application in Non-Destructive Testing

Conventional methods for evaluating the quality of fresh goods are often time consuming, complicated, expensive, and destructive in nature [60]. One of the most significant drawbacks of destructive measurement is the loss of produce after the measurement has been made. To address these limitations, non-destructive optical-based technologies have been developed based on optical measurements in the visible or near-infrared (NIR) regions for quality inspection of various agricultural and food items. Laser Light Backscatter Imaging (LLBI) is a well-known spectrum imaging technology known for its ability to effectively monitor samples without touching them, offering low instrumentation costs and high accuracy [61]. Unlike other imaging techniques, LLBI can obtain spectral data from a sample through deep light penetration. Alternative optical-based imaging techniques have not been used commercially or industrially due to several drawbacks, including subpar performance, adaptability issues, and expensive apparatus, in comparison to LLBI.

Backscattering imaging technology operates by collecting light dispersed after being projected into food substances, as shown in Figure 3. Mireei et al. [62] stated that opaque or semi-transparent substances, such as agricultural products, allow the passage of light at specific wavelengths through their bodies. This phenomenon indicates that 4% of the light is reflected back to the atmosphere when light strikes crop tissue, while the remaining 80% penetrates and is absorbed, transmitted, or scattered back (diffuse reflectance) to the incident location. There are three different types of reflected light: regular, external diffuse, and scattering (Figure 3). The interaction of light with crop tissue during penetration contains valuable information about the material’s structure, crucial for determining yield quality [63,64]. Two significant optical parameters of the backscattering method are the absorption coefficient (a) and the reduced scattering coefficient (s’). The scattering coefficient represents the percentage of light scattered over a material per unit distance, while the absorption coefficient measures the rate at which light intensity decreases as it passes through a food material’s surface [65,66]. Positive interference results from scattered lights with theme-reversed paths; this interference enhances photon projection and reduces the diffusion effect. Enhanced photons are later extracted to quantify the material’s physicochemical properties [67].

The primary components of a LLBI system, depicted in Figure 3 include a computer system, camera, laser-emitting diode, sample platform, supporting frame, and the sample being analyzed. The imaging unit and light source are the two main parts of a LLBI system. A light source generates a continuous light beam, while an imaging device captures a high-quality backscattered image. To produce distortion-free images, the size of the laser beam and the angle at which the sample is incident on the beam must be considered in the light source [68]. The imaging unit primarily consists of a camera; the three most commonly used cameras in the LLBI system are charge-coupled device (CCD), complementary metal oxide (CCMO), and monochromatic [67].

Thousands of spectra can be acquired using the backscattering imaging technique, which is divided into three categories depending on the light source and imaging device used: Laser Light Backscattering Imaging (LLBI), hyperspectral backscattering imaging (HBI), and multispectral backscattering imaging (MBI). These imaging techniques can be distinguished from one another based on the illumination source and imaging setup. The quantification of scattered light is performed at a specific wavelength rather than considering a wide range of wavebands [60].

Figure 3 provides a schematic representation of how laser light distributes in agricultural produce when applied in the Laser Light Backscattering Imaging (LLBI) technique for non-invasive food quality evaluation. It exemplifies three crucial parameters that significantly influence the efficacy of this technique: the laser wavelength, laser beam size, and the incident angle between the food material and the laser source. Adebayo et al. [69] reported that for measuring soluble solids content (SSC) and texture properties, a wavelength ranging from 620 nanometers to 1010 nanometers showed good potential, while moisture content (MC) measurements are best performed at 900 nanometers. Lasers with larger beam sizes are known to have good light distribution but may have poor backscattering properties because photons do not follow the same paths. In contrast, lasers with smaller beam sizes have concentrated light distribution but lower light intensities, which may lead to a poor scattering area. Therefore, choosing the appropriate laser source size is crucial [70]. 

Furthermore, the incident angle between the food material and the laser source must be carefully considered to prevent the camera from receiving irrelevant reflections and reduce the oversaturation effect of light intensity. A review of previous LLBI studies reveals that incident angles between 15° and 22° were used, leading to a simpler image processing process and distortion-free images [69,71].

## 5. Limitations in the Application of Laser in Food Processing 

This industry faces significant challenges as it is cost-sensitive while requiring exceptionally high process throughput. Furthermore, laser-based equipment is sometimes necessary to be positioned in less-than-ideal places due to space constraints and ambient vibration. Paper perforation is just one of the numerous uses in food packaging where laser technology exceeds traditional methods. However, laser and laser system designers must pay close attention to aspects such as system size, cost of ownership, and reliability in order to successfully explore this market. For the industrial application of lasers in food processing, the following limitations must be overcome: laboratory and industrial research on the control of thermal damage to food materials and development of mathematical models, design of automatic and safe processing equipment, consumer acceptance surveys, and the establishment of regulations. 

The food processing industry faces significant challenges due to its cost-sensitive nature and the need for exceptionally high process throughput. To address the limitations in the application of lasers in food processing, various factors need to be considered. Firstly, energy consumption and environmental impact should be taken into account, as lasers can consume a significant amount of energy, which can contribute to higher operational costs and increased carbon emissions. Exploring energy-efficient laser technologies and implementing energy-saving measures can help to address this issue. Secondly, technical expertise and maintenance are crucial for the effective implementation of laser technology in food processing. Companies require skilled personnel with expertise in operating and maintaining laser systems to ensure optimal performance and prevent potential downtime due to system failures. Lastly, regulations and consumer acceptance should also be considered to ensure that laser-based food processing technologies are safe, reliable, and acceptable to consumers. To address these limitations, further research and development are required to improve the efficiency, cost-effectiveness, and safety of laser-based food processing technologies. The advantages and disadvantages of the laser light treatment has been summarized in Table 3.

## 6. Laser Light Technology’s Role in Food Nutrition and Safety 

It has been demonstrated that a variety of food processing activities can be successfully carried out using laser light technology. In the last few decades, a large amount of research has been carried out on the use of laser light to ensure the safety of processed foods. An emerging technology that offers a reliable detection tool for estimating biological samples is the biospeckle laser method. It ensures the excellence and protection of fresh fruit and is quick, easy to use, and affordable. Food such as fruits and vegetables are highly perishable, and handling, transporting, and processing them can cause rapid degradation. Food quality loss is a result of physicochemical and microbiological processes, including variations in surface texture, water content, bruising, browning, and enzymatic alterations, among others [74]. These processes are the principal causes of food deterioration. Laser light that has a wavelength longer than 600 nm is used in biospeckle instruments for illumination or irradiation. Using a biospeckle laser, it is possible to find out more about the biological and physical properties of tissues. As a result, the fields of food quality assurance and control benefit from the use of biospeckle laser technology. It helps in the identification of fruit ripening and maturity changes. It is a valuable tool for spotting ripening, maturing, or bruised fruits and vegetables as well as for quality detection of ayurvedic medicines, meat, and online evaluations of bacterial development, seed viability, biological leaf tissue evaluations, perfusion variation monitoring, and fungal damage [75].

## 7. Future Prospect

We note that the above benefits of laser beams make it possible to suggest the use of laser-assisted processing in the food industry. These laser-assisted food processing experiments should be viewed as exploratory in nature due to the fact that many practical applications still face a number of obstacles and constraints. Since crucial investigations have not yet been concluded, most laser-assisted food processing is still in the laboratory or pilot phase rather than industrial production. It is important to do follow-up research to regulate the heat damage brought on by lasers during food preparation. Therefore, additional research is required to fully understand the production mechanism of novel components formed by the photochemical effect of lasers. Extensive research on laser-processed foods and the gradual improvement of processing information requires a re-evaluation of consumer perceptions of laser-processed foods and the formulation of new rules to govern the industry’s growth in tandem with laser technology.

## 8. Conclusions

In conclusion, laser light technology has enormous potential in the food industry, and its benefits are evident in various applications such as food processing, packaging, and non-destructive testing. Despite some limitations, ongoing research efforts to overcome them and develop cost-effective and efficient laser-based equipment are expected to propel the adoption of this technology. The future scope of laser light technology in food processing lies in the development of optimized equipment for diverse applications, such as laser ablation, microbial inactivation, and food packaging, among others. Future work should focus on enhancing the performance of laser systems while minimizing their environmental impact and energy consumption. Additionally, researchers should continue investigating the optimal frequencies and wavelength for specific applications, developing mathematical models, and conducting consumer acceptance surveys. Further studies are also needed to explore the potential of laser light technology in emerging fields such as precision agriculture and sustainable food production. Overall, laser light technology has the potential to revolutionize the food industry, and its development and application will continue to play a crucial role in enhancing food safety, quality, and sustainability.

## Figures and Tables

**Figure 1 foods-12-02983-f001:**
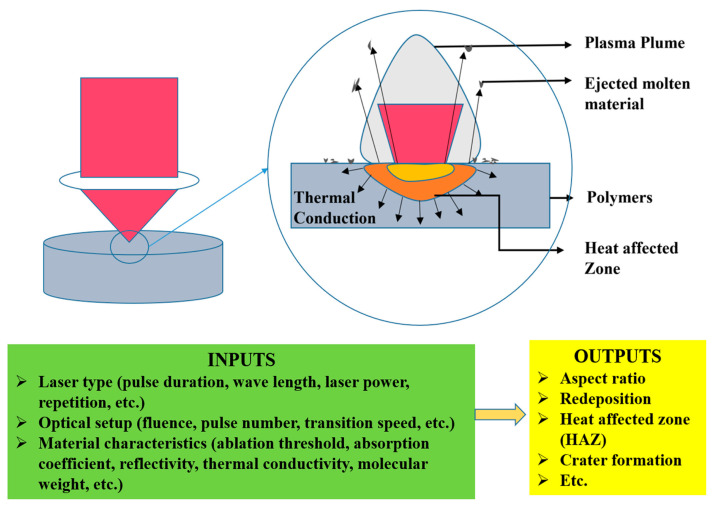
Schematic diagram of the mechanism of laser-induced processing. Reprinted/adapted with permission from [10], 2023, Elsevier.

**Figure 2 foods-12-02983-f002:**
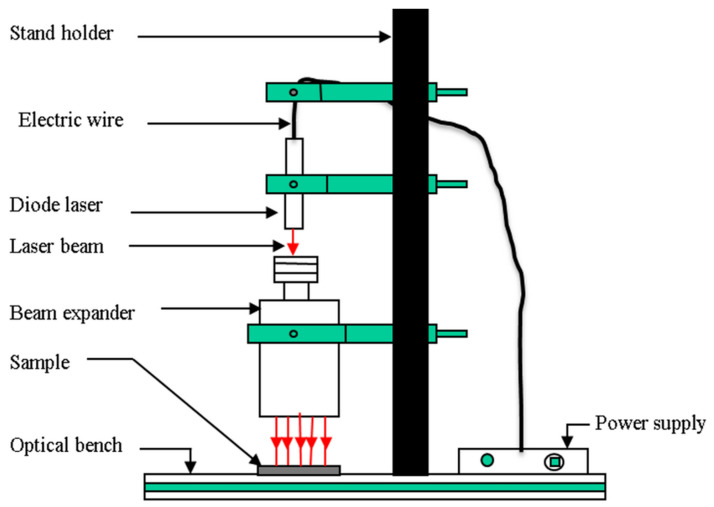
Schematic diagrams of direct laser radiation for microbial inhibition in strawberries. Reprinted/adapted with permission from [25] 2023, Springer.

**Figure 3 foods-12-02983-f003:**
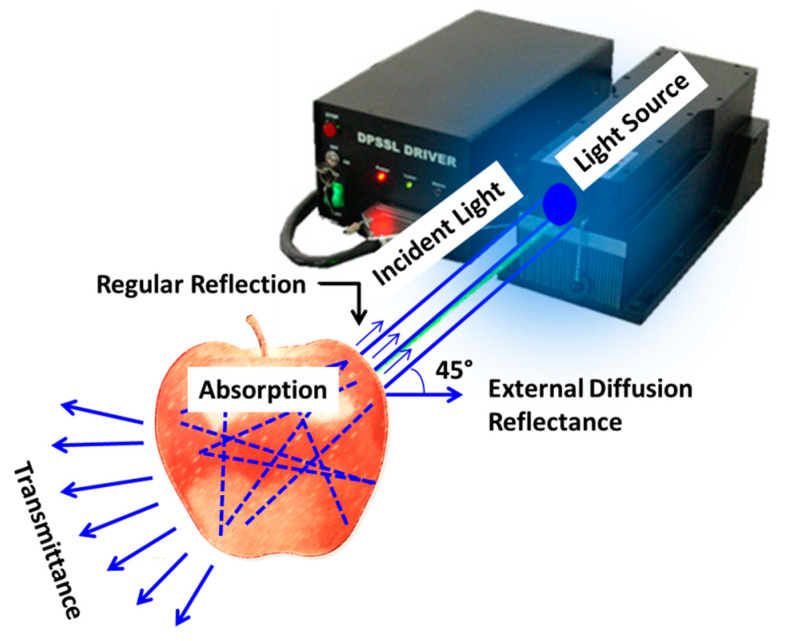
Schematic representation of the distribution of laser light in agricultural produce.

**Table 1 foods-12-02983-t001:** Effect of laser irradiation on microorganisms.

Working Conditions	Food/Microorganism	Inferences	Reference
Nd:YAG laser-355 and 266 nm, energy-185 mJ	Pathogenic bacteria, e.g.,: *P. aeruginosa, Escherichia coli, Staphylococcus aureus, Salmonella typhimurium* and *Listeria monocytogenes*	▪ In baseline assay, *P. aeruginosa* showed maximum inhibition, followed by *Escherichia coli*, Staphylococcus aureus, *Salmonella typhimurium*, and *Listeria monocytogenes* grown on agar surface. ▪ Laser was able to reduce the population of 1.65 × 10^5^ by 4.7 logs, of *P. aeruginosa*. ▪ The reduction patterns of *E. coli*, *Salmonella* sp. Yeasts, and *Lactobacillus* sp. were 30%, 25%, 47%, and 30%, respectively, with laser technology.	[28]
Mid-IR femtosecond laser radiation central wavelength: 5.8 and 3.4 μm pulse duration τ ≈ 130 f pulse energies of 10 μJ (6 μm) and 30 μJ (3 μm) repetition rate of 1 kHz	*Pseudomonas aeruginosa* bacteria	▪ The bacterial inactivation by the 6 μm ultrashort laser pulses is attributed to dissociative denaturation of lipids and proteins in the cell membranes and intra-cell nucleic acids.	[29]
Nd:YAG and Er laser-single pulse train mode Pulse energies-40–400 mJ Pulse lengths of 100–1000 μs Output power-15 W Frequency-100 Hz. Time-5–120 s.	*Enterococcus faecalis, Candida albicans* or *Propionibacterium acnes*	▪ Thresholds were around 5300 J/cm^2^ for *C. albicans* and 7100 J/cm^2^ for *P. acnes*. ▪ No inhibition was observed for *E. faecalis*. ▪ Er:YAG irradiation was superior to Nd:YAG in inactivating microorganisms.	[30]
445 nm and 650 nm laser activated the chlorophyll extract of the papaya leaf (0.5 mg/L)	*Candida albicans*	▪ The absorbance percentage of chlorophyll extracts on wavelengths of 650 nm and 445 nm, respectively, were 22.26% and 60.29%, respectively. ▪ The inactivation was about 32% for 650 nm (*p* = 0.001), while the 445 nm lasers only 25% (*p* = 0.061). ▪ The maximum malondialdehyde levels by treatment of the laser 650 nm were (0.046 ± 0.004) nmol/mg.	[31]
Michelia oil (Michelia alba) (linalool and caryophyllene) (300–500 mg g^−1^) Helium-Neon (He-Ne) operating at 543, 594, 604, 612, and 633 nm for 1 min Storage: 8 weeks at 25 °C and 100% RH brown rice snack bars	*Aspergillus niger, Aspergillus flavus, Penicillium* sp., *Rhizopus* sp., *Fusarium* sp. and *Cladosporium* sp.	▪ The minimum inhibitory concentrations of michelia oil and linalool with laser treatment were 300 mg g^−1^ and 150 mg g^−1^, respectively. Without laser treatment, no MICs were found at the maximum concentration (500 mg g^−1^). ▪ Shelf life of laser treated samples were also 4 days more than the control samples during storage under the accelerated conditions.	[32]
Nd: YAG laser—1064 nm Power—10–50 Watt Time, 2 min	Food—raw cow milk	▪ Complete sterilization of bacteria and fungi at laser power 50 W for 2 min. ▪ pH of milk was reduced from 6.18 for the control sample in the first day down to 4.50 for the highest laser-treated sample after five days. ▪ pH of milk was reduced when increasing laser output power.	[33]

**Table 2 foods-12-02983-t002:** Applications of laser in food packaging technology.

Type of Laser	Packaging Material	Results	Reference
Laser transmission welding—low power pulsed wave fibre laser	Polylactide and aluminium	An accurate relationship between the joint quality and both the welding speed and the k-factor, which represents the delivered energy per unit length and affects the bonding mechanism at the interface, was determined. The achieved feasibility area is extremely narrow and possible only for the higher value of the average power.	[49]
Femtosecond laser processing and hot embossing technique Two types of topographies for Lubricating Intensive Surfaces (LIS)-single-scale sub-micron laser-induced periodic surface structures (LIPSS), multi-scale (MS) structures with both micron and sub-micron features	Polypropylene, polystyrene, stainless steel on water, milk, and honey	The critical sliding angles at which liquid droplets attained motion on LIS were observed to be less than 32° for all investigated liquids. The LIPSS-LIS substrates retained their functionality even after subjecting them to severe vibration. Both vibration- and shear-induced loss of lubricant impacted the MS-LIS functionality.	[50]
Laser transmission welding	Polyethylene to polypropylene substrate	Parameters required for transmission were optimized on the basis of numerical simulations with defined bonded track widths.	[51]
Laser-etched pouches (gas control functions)	Kimchi	The concentration of carbon dioxide in the pouches with a high gas transmission rate was less than that in other pouches (*p* < 0.05), indicating that low a carbon dioxide concentration resulted in less volume expansion. Few differences were observed between the quality characteristics of kimchi (for example, pH, titratable acidity, organic acid, and microbial count). The use of laser-etched films could control gas inside packages and exert significant effects on alleviating the volume expansion or pressure build-up in kimchi packages.	[52]
High-power diode laser joining Laser transmission welding	Aluminum films coated with a polyester resin with polypropylene (PP) films	Analysis of the mechanical response of the welded joints allowed to identify the optimal processing window, that is, the choice of the operational parameters that leads to satisfactory welded joints, stating the high potential of laser systems in the joining process of aluminum and PP films for food packaging applications.	[53]
Thermal activation using 27 W CO_2_ laser	Polypropylene substrate/adhesive coating	The adhesion between the material combination trialled here responded linearly to thermal load. The processing window of an incident CO_2_ laser spot increases with respect to spot diameter, thus yielding greater bond stability in the face of short-term laser variance. Novel empirical tool is developed that predicts the CO_2_ laser power required to achieve a viable adhesive bond for this material combination. This will enable the packaging industry to achieve markedly increased financial yield, process efficiency, reduced material waste, and process flexibility.	[54]
Laser ablation	Laminated aluminum (Al) film on the parchment paper substrate	The intrinsic humidity-responsive characteristics of the laser-induced Al_2_O_3_ nanostructures provide the wireless sensor with a tenfold higher sensitivity to humidity than a similar LC resonant sensor prepared by conventional photolithography-based processes on FR-4 substrates. The frequency change of the sensor is observed to be a linear function within the range of 0−85% RH, providing an average sensitivity of −87 kHz RH^−1^ with good repeatability and stable performance.	[55]
The microperforation using a pulsed fibre laser technique Breathable polymeric packaging films—laser power-20 W pulse duration 200 ns. The numbers of holes (80 μm in diameter) 2000 holes/m^2^ and 4000 holes/m^2^.	Polymeric films	The number of perforations affected the water vapour transmission (WVT): the WVT was 11 g/m^2^/d for unperforated film, and 60 g/m^2^/d for the film with 4000 holes/m^2^, indicating that the fibre laser can be used successfully for micro perforation.	[56]
Polyethylene terephthalate/polyethylene laminated film perforated Carbon dioxide (CO_2_) laser Infrared wavelength of 10.2 μm Pulse duration 3 to 200 μs Threshold PET −74.0 J/cm^2^, PE −92.5 J/cm^2^.	Mixed vegetable salad	The laser perforation on the PET side used up to 50% less energy than the perforation from the PE side to create the same size through-microhole. The steady state level of gas composition of 10–13% O_2_ and 8–10% CO_2_ was achieved in perforated packages that could keep the mixed salad fresh for 9 days at 8 °C.	[57]

**Table 3 foods-12-02983-t003:** Applications, advantages, and disadvantages of laser light treatment in various food sectors and associated potential issues [36,39,72,73].

Application in Food Sector	Advantages	Disadvantages	Potential Issues
Laser cooking for precise and controlled heat application.	Lasers may supply the same heat for cooking with the best possible control, reproducibility for targeted energy, and the highest resolution.	It may thermally damage the food materials.	Risk of overcooking or undercooking due to high precision requirements.
High-resolution heat processing in various food products	High-resolution heating is possible with lasers, suitable for a wide variety of food applications.	It has a significant initial capital cost of equipment.	High upfront costs and maintenance expenditures.
Use in sterilization and decontamination processes.	Lasers may develop minimal contamination in the processed food.	Low efficiency of lasers.	Lower overall energy efficiency compared to conventional methods.
Used in non-contact cutting or engraving processes in food preparation.	The noncontact nature of lasers helps to maintain the quality of the final processed product.	When the laser is not continually employed, energy is wasted via beam dumping.	Inefficient energy usage in intermittent processes
Precision cutting in meat or fish industries.	The primary benefit of laser light is the ability to control beam power by adjusting the current flowing through the electric discharge.	A small heat-impacted zone will form along the cut edge of parts heated during high intensity laser processing.	Changes in product quality at the cut edges.
Metal-free packaging cutting and engraving	Laser light has the benefit of having a minimum distortion and heat impacted zone.	High laser beam reflectivity on metals. All metals cannot be cut with a laser beam due to issues with beam reflections.	Limitations with metal-containing packaging materials.
Emerging applications in custom 3D-printed foods.	Laser technology allows for the potential of 3D food printing.	The technology is still in the experimental stage and not yet widely adopted.	Consumer acceptance and regulatory considerations for 3D-printed foods.
Quality assurance and process control in food industries	Lasers can assist in the monitoring and control of food processing parameters.	Requires complex sensor systems and data processing.	Implementation complexity and cost of sensor systems and data processing infrastructure.

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
