# Peer review of "Laser Light as an Emerging Method for Sustainable Food Processing, Packaging, and Testing"

_foods, 2023, doi:10.3390/foods12162983_

Round 1
Reviewer 1 Report
The idea of the current review entitled “Laser Light as an Emerging Method for Sustainable Food Processing, Packaging, and Testing” is interesting. Please check the following comments:
Abstract:
Please include one/two sentence(s) about what will be covered in this review.
Introduction:
At the end of the introduction, you can add some highlights about the fields of food technology where Laser light technology seems promising, including some references.
You should also highlight the topics that will be covered in this review.
Lone 103: Please add reference(s) after ……..Beer-Lambert's law ().
Line 212: the microbes' name should be italicized wherever applied. Please check the entire manuscript for this.
A separate section on the inactivation mechanism by laser radiation is needed to better understand the process.
Figure 3: the reference of the caption should be similar to the rest of the citation style.
Some superscripts/subscripts were not properly shown, for example- m2, CO2; please check the entire manuscript.
Check the problems with the page number after page 6.
There are two Table 1!
Citation styles are not uniform throughout the manuscript. Recheck all of them.
General Comments:
-A section of current regulations used for laser radiation should be added.
-A section on prospect/possibilities/future research should be added.
-If the present title of the review exists, a section on how laser radiation will help in sustainability should be included. There should be some information in the introduction too.
Overall Comments:
I found 37% plagiarism in this manuscript (without references), with up to 6% from a single source. Several sections should be rewritten to check this.
There are many typos and grammatical in the manuscript. Authors are highly encouraged to revise the paper to check these.
Reviewer 2 Report
The authors have compiled excellent summary on Laser-assisted technology for sustainable food Processing, Packaging, and Testing.
There are several reports already available, published by other researcher (https://doi.org/10.1016/j.tifs.2021.10.031; 10.1016/j.ifset.2011.02.008; e.t.c.). So, major concern is to prove how the present review is more useful and updated to readers, compared other which are already published.
Moreover, their ares one concern that authors needs to reflect before the manuscript can be accepted.
Authors suggested to add Challenges and future perspectives of various technology for sustainable food Processing, Packaging, and Testing.
Suggested to add a comparative data tabulated for different lasers used in food preservation.
Line no. 173, 198. Write complete name of pathogen
Line no. 222, 225, 226, 228, 230, 231, . Italic the pathogen name
Line 227, 234. Please correct the unit
The resolution of figure is not up to the journal standard
Suggested to add drawbacks of the laser assisted technology.
Though technology advanced the laser assisted preservation of food, their are potential side effects of such technology, authors are suggested to summarise such aspects too.
Thanks and good luck
Minor editing of English language required
Reviewer 3 Report
The present study attempts to provide further insight on the laser light technique used as a novel and sustainable method for food processing and packaging in the agri-food industry.
As this subject is of interest for the readers of the Foods Journal, below is a list of potential things that could be improved.
(1) The Introduction subchapter need improvements, more research studies regarding the use of laser light technology use in the agri-food industry need to be added/cited in order to give a more in-depth analysis of the literature review and other findings found by other authors.
(2) The authors need to follow the guidelines for citing in text of Foods.
(3) In table 1 all microbial strains need to be written in italic
(4) There are several places in the paper where CO2 (eg. line 381) and other chemical formulas are not written correct.
(5) When discussing novel processing techniques like lase light treatment, there are few aspects that should be clarified, starting with applicability, and the effect on the shelf life, safety, and nutritional value of the food products exposed to the treatment. The first two was discussed in the review; it would be interesting to see the last two.
(6) A table consisting advantages and disadvantages of the laser light treatment should be included in order to provide a more in-depth view over the potential use of this method.
The paper is written in a good manner, only minor corrections are needed.
Round 2
Reviewer 1 Report
The authors responded to most of the comments. However, some points need to be taken care of as follows:
-There are still some typos, punctuation, and grammatical errors after revision. Please revise the manuscript thoroughly by an expert academic editor to solve these.
-Table 1: Authors yet need to change the microbes’ names in italic form under the column “Inferences”. Again, check the entire manuscript for this.
-The information mentioned in Section 5 and Table 3 could be more specific. -You need particular information with proper citations to use this kind of information in a review/research paper. I recommend merging Table 3 in writings with appropriate sources.
Minor editing of the English language required
Reviewer 3 Report
The authors managed to improve the article by addressing the pointed issues, thus answering in a professional manner and adding the necessary improvements in order for the article to highlight the importance of the discussed topic.
The quality of English is good, only minor changes are needed.
